# FLamby: Datasets and Benchmarks for Cross-Silo Federated Learning in Realistic Healthcare Settings

**Jean Ogier du Terrail**[1]   **Samy-Safwan Ayed**[2]   **Edwige Cyffers**[3]   **Felix Grimberg**[4]
**Chaoyang He**[5]   **Regis Loeb**[1]   **Paul Mangold**[3]   **Tanguy Marchand**[1]
**Othmane Marfoq**[2]   **Erum Mushtaq**[6]   **Boris Muzellec**[1]   **Constantin Philippenko**[7]
**Santiago Silva**[2]   **Maria Teleńczuk**[1]   **Shadi Albarqouni**[8,9] **Salman Avestimehr**[5,6]
**Aurélien Bellet**[3]   **Aymeric Dieuleveut**[7]   **Martin Jaggi**[4]
**Sai Praneeth Karimireddy**[10] **Marco Lorenzi**[2]   **Giovanni Neglia**[2]   **Marc Tommasi**[3]
**Mathieu Andreux**[1]

[1]Owkin, Inc, [2] Inria, Université Côte d'Azur, Sophia Antipolis, France
[3]Univ. Lille, Inria, CNRS, Centrale Lille, UMR 9189 - CRIStAL, F-59000 Lille, France
[4]EPFL   [5]FedML, Inc.   [6]University of Southern California
[7]CMAP, UMR 7641, École Polytechnique, Institut Polytechnique de Paris
[8]University Hospital Bonn   [9]Helmholtz Munich   [10]University of California, Berkeley
{jean.du-terrail, regis.loeb, tanguy.marchand, boris.muzellec,
maria.telenczuk, mathieu.andreux}@owkin.com, {samy-safwan.ayed,
edwige.cyffers, paul.mangold, othamne.marfoq,
santiago-smith.silva-rincon, aurelien.bellet,
marco.lorenzi, giovanni.neglia, marc.tommasi}@inria.fr
{felix.grimberg, martin.jaggi}@epfl.ch,
ch@fedml.ai, {emushtaq, avestime}@usc.edu,
{constantin.philippenko, aymeric.dieuleveut}@polytechnique.edu,
shadi.albarqouni@ukbonn.de, sp.karimireddy@berkeley.edu

## Abstract

Federated Learning (FL) is a novel approach enabling several clients holding sensitive data to collaboratively train machine learning models, without centralizing data. The cross-silo FL setting corresponds to the case of few (2–50) reliable clients, each holding medium to large datasets, and is typically found in applications such as healthcare, finance, or industry. While previous works have proposed representative datasets for cross-device FL, few realistic healthcare cross-silo FL datasets exist, thereby slowing algorithmic research in this critical application. In this work, we propose a novel cross-silo dataset suite focused on healthcare, FLamby (Federated Learning AMple Benchmark of Your cross-silo strategies), to bridge the gap between theory and practice of cross-silo FL. FLamby encompasses 7 healthcare datasets with natural splits, covering multiple tasks, modalities, and data volumes, each accompanied with baseline training code. As an illustration, we additionally benchmark standard FL algorithms on all datasets. Our flexible and modular suite allows researchers to easily download datasets, reproduce results and re-use the different components for their research. FLamby is available at `www.github.com/owkin/flamby`.

## 1   Introduction

Recently it has become clear that, in many application fields, impressive machine learning (ML) task performance can be reached by scaling the size of both ML models and their training data while

36th Conference on Neural Information Processing Systems (NeurIPS 2022) Track on Datasets and Benchmarks.

keeping existing well-performing architectures mostly unaltered [118, 76, 24, 109]. In this context, it is often assumed that massive training datasets can be collected and centralized in a single client in order to maximize performance. However, in many application domains, data collection occurs in distinct sites (further referred to as clients, e.g., mobile devices or hospitals), and the resulting local datasets cannot be shared with a central repository or data center due to privacy or strategic concerns [42, 18].

To enable cooperation among clients given such constraints, Federated Learning (FL) [99, 73] has emerged as a viable alternative to train models across data providers without sharing sensitive data. While initially developed to enable training across a large number of small clients, such as smartphones or Internet of Things (IoT) devices, it has been then extended to the collaboration of fewer and larger clients, such as banks or hospitals. The two settings are now respectively referred to as *cross-device* FL and *cross-silo* FL, each associated with specific use cases and challenges [73].

On the one hand, cross-device FL leverages edge devices such as mobile phones and wearable technologies to exploit data distributed over billions of data sources [99, 16, 14, 103]. Therefore, it often requires solving problems related to edge computing [53, 87, 129], participant selection [73, 131, 23, 44], system heterogeneity [73], and communication constraints such as low network bandwidth and high latency [113, 93, 51]. On the other hand, cross-silo initiatives enable to untap the potential of large datasets previously out of reach. This is especially true in healthcare, where the emergence of federated networks of private and public actors [112, 115, 105], for the first time, allows scientists to gather enough data to tackle open questions on poorly understood diseases such as triple negative breast cancer [40] or COVID-19 [34]. In cross-silo applications, each silo has large computational power, a relatively high bandwidth, and a stable network connection, allowing it to participate to the whole training phase. However, cross-silo FL is typically characterized by high inter-client dataset heterogeneity and biases of various types across the clients [105, 40].

As we show in Section 2, publicly available datasets for the cross-silo FL setting are scarce. As a consequence, researchers usually rely on heuristics to artificially generate heterogeneous data partitions from a single dataset and assign them to hypothetical clients. Such heuristics might fall short of replicating the complexity of natural heterogeneity found in real-world datasets. The example of digital histopathology [126], a crucial data type in cancer research, illustrates the potential limitations of such synthetic partition methods. In digital histopathology, tissue samples are extracted from patients, stained, and finally digitized. In this process, known factors of data heterogeneity across hospitals include patient demographics, staining techniques, storage methodologies of the physical slides, and digitization processes [71, 45, 59]. Although staining normalization [81, 35] has seen recent progress, mitigating this source of heterogeneity, the other highlighted sources of heterogeneity are difficult to replicate with synthetic partitioning [59] and some may be unknown, which calls for actual cross-silo cohort experiments. This observation is also valid for many other application domains, e.g. radiology [52], dermatology [10], retinal images [10] and more generally computer vision [122].

In order to address the lack of realistic cross-silo datasets, we propose FLamby, an open source cross-silo federated dataset suite with natural partitions focused on healthcare, accompanied by code examples, and benchmarking guidelines. Our ambition is that FLamby becomes the reference benchmark for cross-silo FL, as LEAF [19] is for cross-device FL. To the best of our knowledge, apart from some promising isolated works to build realistic cross-silo FL datasets (see Section 2), our work is the first standard benchmark allowing to systematically study healthcare cross-silo FL on different data modalities and tasks.

To summarize, our contributions are threefold:

1. We build an open-source federated cross-silo healthcare dataset suite including 7 datasets. These datasets cover different tasks (classification / segmentation / survival) in multiple application domains and with different data modalities and scale. Crucially, all datasets are partitioned using natural splits.

2. We provide guidelines to help compare FL strategies in a fair and reproducible manner, and provide illustrative results for this benchmark.

3. We make open-source code accessible for benchmark reproducibility and easy integration in different FL frameworks, but also to allow the research community to contribute to FLamby development, by adding more datasets, benchmarking types and FL strategies.

Table 1: Overview of the datasets, tasks, metrics and baseline models in FLamby. For Fed-Camelyon16 the two different sizes refer to the size of the dataset before and after tiling.

| Dataset | Fed-Camelyon16 | Fed-LIDC-IDRI | Fed-IXI | Fed-TCGA-BRCA | Fed-KITS2019 | Fed-ISIC2019 | Fed-Heart-Disease |
|---|---|---|---|---|---|---|---|
| Input (x) | Slides | CT-scans | T1WI | Patient info. | CT-scans | Dermoscopy | Patient info. |
| Preprocessing | Matter extraction + tiling | Patch Sampling | Registration | None | Patch Sampling | Various image transforms | Removing missing data |
| Task type | binary classification | 3D segmentation | 3D segmentation | survival | 3D segmentation | multi-class classification | binary classification |
| Prediction (y) | Tumor on slide | Lung Nodule Mask | Brain mask | Risk of death | Kidney and tumor masks | Melanoma class | Heart disease |
| Center extraction | Hospital | Scanner Manufacturer | Hospital | Group of Hospitals | Group of Hospitals | Hospital | Hospital |
| Thumbnails | | | | | | | |
| Original paper | Litjens *et al.* 2018 | Armato *et al.* 2011 | Perez *et al.* 2021 | Liu *et al.* 2018 | Heller *et al.* 2019 | Tschandl *et al.* 2018 / Codella *et al.* 2017 / Combalia *et al.* 2019 | Janosi *et al.* 1988 |
| # clients | 2 | 5 | 3 | 5 | 6 | 5 | 4 |
| # examples | 399 | 1,018 | 566 | 1,088 | 96 | 23,247 | 740 |
| # examples per center | 239, 150 | 670, 205, 69, 74 | 311, 181, 74 | 311, 196, 206, 162, 51 | 12, 14, 12, 12, 16, 30 | 12413, 3954, 3363, 2259, 819, 439 | 303, 261, 46, 130 |
| Model | DeepMIL [66] | Vnet [100, 102] | 3D U-net [25] | Cox Model [33] | nnU-Net [69] | efficientnet [119] + linear layer | Logistic Regression |
| Metric | AUC | DICE | DICE | C-index | DICE | Balanced Accuracy | Accuracy |
| Size | 50G (850G total) | 115G | 444M | 115K | 54G | 9G | 40K |
| Image resolution | 0.5 μm / pixel | ∼1.0 × 1.0 × 1.0 mm / voxel | ∼1.0 × 1.0 × 1.0 mm / voxel | NA | ∼1.0 × 1.0 × 1.0 mm / voxel | ∼0.02 mm / pixel | NA |
| Input dimension | 10,000 x 2048 | 128 x 128 x 128 | 48 x 60 x 48 | 39 | 64 x 192 x 192 | 200 x 200 x 3 | 13 |

This paper is organized as follows. Section 2 reviews existing FL datasets and benchmarks, as well as client partition methods used to artificially introduce data heterogeneity. In Section 3, we describe our dataset suite in detail, notably its structure and the intrinsic heterogeneity of each federated dataset. Finally, we define a benchmark of several FL strategies on all datasets and provide results thereof in Section 4.

## 2  Related Work

In FL, data is collected locally in clients in different conditions and without coordination. As a consequence, clients' datasets differ both in size (unbalanced) and in distribution (non-IID) [99]. The resulting *statistical heterogeneity* is a fundamental challenge in FL [84, 73], and it is necessary to take it into consideration when evaluating FL algorithms. Most FL papers simulate statistical heterogeneity by artificially partitioning classic datasets, e.g., CIFAR-10/100 [80], MNIST [83] or ImageNet [37], on a given number of clients. Common approaches to produce synthetic partitions of classification datasets include associating samples from a limited number of classes to each client [99], Dirichlet sampling on the class labels [61, 133], and using Pachinko Allocation Method (PAM) [86, 110] (which is only possible when the labels have a hierarchical structure). In the case of regression tasks, [107] partitions the *superconduct* dataset [20] across 20 clients using Gaussian Mixture clustering based on T-SNE representations [124] of the features. Such synthetic partition approaches may fall short of modelling the complex statistical heterogeneity of real federated datasets. Evaluating FL strategies on datasets with natural client splits is a safer approach to ensuring that new strategies address real-world issues.

For *cross-device* FL, the LEAF dataset suite [19] includes five datasets with natural partition, spanning a wide range of machine learning tasks: natural language modeling (Reddit [127]), next character prediction (Shakespeare [99]), sentiment analysis (Sent140 [47]), image classification (CelebA [90]) and handwritten-character recognition (FEMNIST [28]). TensorFlow Federated [15] complements LEAF and provides three additional naturally split federated benchmarks, i.e., StackOverflow [120], Google Landmark v2 [62] and iNaturalist [125]. Further, FLSim [111] provides cross-device examples based on LEAF and CIFAR10 [80] with a synthetic split, and FedScale [82] introduces a large FL benchmark focused on mobile applications. Apart from iNaturalist, the aforementioned datasets target the cross-device setting.

To the best of our knowledge, no extensive benchmark with natural splits is available for *cross-silo* FL. However, some standalone works built cross-silo datasets with real partitions. [48] and [97] partition Cityscapes [30] and iNaturalist [125], respectively, exploiting the geolocation of the picture

acquisition site. [60] releases a real-world, geo-tagged dataset of common mammals on Flickr. [94] gathers a federated cross-silo benchmark for object detection created using street cameras. [31] partitions Vehicle Sensor Dataset [41] and Human Activity Recognition dataset [7] by sensor and by individuals, respectively. [95] builds an iris recognition federated dataset across five clients using multiple iris datasets [128, 135, 136, 108]. While FedML [55] introduces several cross-silo benchmarks [56, 132, 54], the related client splits are synthetically obtained with Dirichlet sampling and not based on a natural split. Similarly, FATE [1] provides several cross-silo examples but, to the best of our knowledge, none of them stems from a natural split.

In the medical domain, several works use natural splits replicating the data collection process in different hospitals: the works [5, 21, 11, 74, 130, 22] respectively use the Camelyon datasets [89, 13, 12], the CheXpert dataset [67], LIDC dataset [8], the chest X-ray dataset [78], the IXI dataset [130], the Kaggle diabetic retinopathy detection dataset [49]. Finally, the works [6, 50, 91] use the TCGA dataset [121] by extracting the Tissue Source site metadata.

Our work aims to give more visibility to such isolated cross-silo initiatives by regrouping seven medical datasets, some of which listed above, in a single benchmark suite. We also provide reproducible code alongside precise benchmarking guidelines in order to connect past and subsequent works for a better monitoring of the progress in cross-silo FL.

# 3   The FLamby Dataset Suite

## 3.1   Structure Overview

The FLamby datasets suite is a Python library organized in two main parts: datasets with corresponding baseline models, and FL strategies with associated benchmarking code. The suite is modular, with a standardized simple application programming interface (API) for each component, enabling easy re-use and extensions of different components. Further, the suite is compatible with existing FL software libraries, such as FedML [55], Fed-BioMed [117], or Substra [46]. Listing 1 provides a code example of how the structure of FLamby allows to test new datasets and strategies in a few lines of code, and Table 1 provides an overview of the FLamby datasets.

**Dataset and baseline model.**   The FLamby suite contains datasets with a natural notion of client split, as well as a predefined task and associated metric. A train/test set is predefined for each client to enable reproducible comparisons. We further provide a baseline model for each task, with a reference implementation for training on pooled data. For each dataset, the suite provides documentation, metadata and helper functions to: 1. download the original pooled dataset; 2. apply preprocessing if required, making it suitable for ML training; 3. split each original pooled dataset between its natural clients; and 4. easily iterate over the preprocessed dataset. The dataset API relies on PyTorch [104], which makes it easy to iterate over the dataset with natural splits as well as to modify these splits if needed.

**FL strategies and benchmark.**   FL training algorithms, called *strategies* in the FLamby suite, are provided for simulation purposes. In order to be agnostic to existing FL libraries, these strategies are provided in plain Python code. The API of these strategies is standardized and compatible with the dataset API, making it easy to benchmark each strategy on each dataset. We further provide a script performing such a benchmark for illustration purposes. We stress the fact that it is easy to alternatively use implementations from existing FL libraries.

## 3.2   Datasets, Metrics and Baseline Models

We provide a brief description of each dataset in the FLamby dataset suite, which is summarized in Table 1. In Section 3.4, we further explore the heterogeneity of each dataset, as displayed in Figure 1.

**Fed-Camelyon16.**   Camelyon16 [89] is a histopathology dataset of 399 digitized breast biopsies' slides with or without tumor collected from two hospitals: Radboud University Medical Center (RUMC) and University Medical Center Utrecht (UMCU). By recovering the original split information we build a federated version of Camelyon16 with **2** clients. The task consists in binary classification

of each slide, which is challenging due to the large size of each image ($10^5 \times 10^5$ pixels at 20X magnification), and measured by the Area Under the ROC curve (AUC).

As a baseline, we follow a weakly-supervised learning approach. Slides are first converted to bags of local features, which are one order of magnitude smaller in terms of memory requirements, and a model is then trained on top of this representation. For each slide, we detect regions with a matter-detection network and then extract features from each tile with an ImageNet-pretrained Resnet50, following state-of-the-art practice [32, 92]. Note that due to the imbalanced distribution of tissue in the different slides, a different number of features is produced for each slide: we cap the total number of tiles to $10^5$ and use zero-padding for consistency. We then train a DeepMIL architecture [65], using its reference implementation [66] and hyperparameters from [36]. We refer to Appendix C for more details.

**Fed-LIDC-IDRI.**  LIDC-IDRI [8, 64, 26] is an image database [26] study with 1018 CT-scans (3D images) from The Cancer Imaging Archive (TCIA), proposed in the LUNA16 competition [114]. The task consists in automatically segmenting lung nodules in CT-scans, as measured by the DICE score [39]. It is challenging because lung nodules are small, blurry, and hard to detect. By parsing the metadata of the CT-scans from the provided annotations, we recover the manufacturer of each scanning machine used, which we use as a proxy for a client. We therefore build a **4**-client federated version of this dataset, split by manufacturer. Figure 1b displays the distribution of voxel intensities in each client.

As a baseline model, we use a VNet [100] following the implementation from [102]. This model is trained by sampling 3D-volumes into 3D patches fitting in GPU memory. Details of the sampling procedure are available in Appendix D.

**Fed-IXI.**  This dataset is extracted from the Information eXtraction from Images - IXI database [38], and has been previously released by Perez *et al.* [2, 106] under the name of *IXITiny*. IXITiny provides a database of brain T1 magnetic resonance images (MRIs) from **3** hospitals (Guys, HH, and IOP). This dataset has been adapted to a brain segmentation task by obtaining spatial brain masks using a state-of-the-art unsupervised brain segmentation tool [63]. The quality of the resulting supervised segmentation task is measured by the DICE score [39].

The image pre-processing pipeline includes volume resizing to $48 \times 60 \times 48$ voxels, and sample-wise intensity normalization. Figure 1c highlights the heterogeneity of the raw MRI intensity distributions between clients. As a baseline, we use a 3D U-net [25] following the implementation of [3]. Appendix E provides more detailed information about this dataset, including demographic information, and about the baseline.

**Fed-TCGA-BRCA.**  The Cancer Genome Atlas (TCGA)'s Genomics Data Commons (GDC) portal [101] contains multi-modal data (tabular, 2D and 3D images) on a variety of cancers collected in many different hospitals. Here, we focus on clinical data from the BReast CAncer study (BRCA), which includes features gathered from 1066 patients. We use the Tissue Source Site metadata to split data based on extraction site, grouped into geographic regions to obtain large enough clients. We end up with **6** clients: USA (Northeast, South, Middlewest, West), Canada and Europe, with patient counts varying from 51 to 311. The task consists in predicting survival outcomes [72] based on the patients' tabular data (39 features overall), with the event to predict being death. This survival task is akin to a ranking problem with the score of each sample being known either directly or only by lower bound (right censorship). The ranking is evaluated by using the concordance index (C-index) that measures the percentage of correctly ranked pairs while taking censorship into account.

As a baseline, we use a linear Cox proportional hazard model [33] to predict time-to-death for patients. Figure 1e highlights the survival distribution heterogeneity between the different clients. Appendix F provides more details on this dataset.

**Fed-KITS2019.**  The KiTS19 dataset [57, 58] stems from the Kidney Tumor Segmentation Challenge 2019 and contains CT scans of 210 patients along with the segmentation masks from 79 hospitals. We recover the hospital metadata and extract a **6**-client federated version of this dataset by removing hospitals with less than 10 training samples. The task consists of both kidney and tumor segmentation, labeled 1 and 2, respectively, and we measure the average of Kidney and Tumor DICE scores [39] as our evaluation metric.

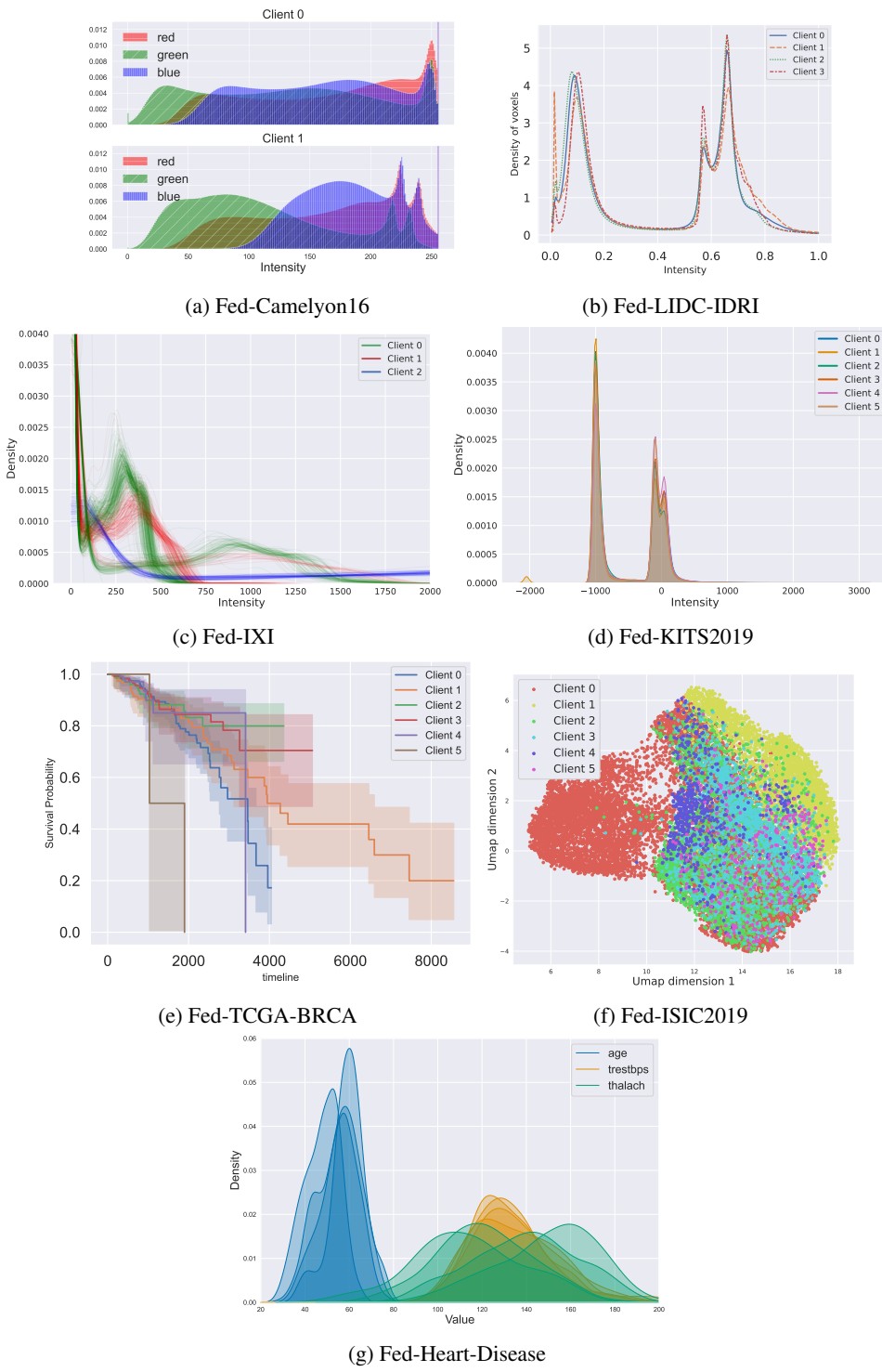

(a) Fed-Camelyon16

(b) Fed-LIDC-IDRI

(c) Fed-IXI

(d) Fed-KITS2019

(e) Fed-TCGA-BRCA

(f) Fed-ISIC2019

(g) Fed-Heart-Disease

Figure 1: Heterogeneity of FLamby datasets. Best seen in color. 1a: Color histograms per client. 1b, 1c and 1d: Voxel intensity distribution per client. 1e: Kaplan-Meier survival curves per client. 1f: UMAP of deep network features of the raw images, colored by client. 1g: Per-client histograms of several features. Differences between client distributions are sometimes obvious and sometimes subtle. Some clients are close in the feature space, some are not and different types of heterogeneity are observed with different data modalities.

The preprocessing pipeline comprises intensity clipping followed by intensity normalization, and resampling of all the cases to a common voxel spacing of 2.90x1.45x1.45 mm. As a baseline, we use the nn-Unet library [69] to train a 3D nnU-Net, combined with multiple data augmentations including scaling, rotations, brightness, contrast, gamma and Gaussian noise with the batch generators framework [68]. Appendix G provides more details on this dataset.

**Fed-ISIC2019.** The ISIC2019 dataset [123, 27, 29] contains dermoscopy images collected in 4 hospitals. We restrict ourselves to 23,247 images from the public train set due to metadata availability reasons, which we re-split into train and test sets. The task consists in image classification among 8 different melanoma classes, with high label imbalance (prevalence ranging 49% to less than 1% depending on the class). We split this dataset based on the imaging acquisition system used: as one hospital used 3 different imaging technologies throughout time, we end up with a **6**-client federated version of ISIC2019. We measure classification performance through balanced accuracy, defined as the average recall on each class.

As an offline preprocessing step, we follow recommendations and code from [9] by resizing images to the same shorter side while maintaining their aspect ratio, and by normalizing images' brightness and contrast through a color consistency algorithm. As a baseline classification model, we fine-tune an EfficientNet [119] pretrained on ImageNet, with a weighted focal loss [88] and with multiple data augmentations. Figure 1f highlights the heterogeneity between the different clients prior to preprocessing. Appendix H provides more details on this dataset.

**Fed-Heart-Disease.** The Heart-Disease dataset [70] was collected in **4** hospitals in the USA, Switzerland and Hungary. This dataset contains tabular information about 740 patients distributed among these four clients. The task consists in binary classification to assess the presence or absence of heart disease. We preprocess the dataset by removing missing values and encoding non-binary categorical variables as dummy variables, which gives 13 relevant attributes. As a baseline model, we use logistic regression. Appendix I provides more details on this dataset.

## 3.3 Federated Learning Strategies in FLamby

The following standard FL algorithms, called *strategies*, are implemented in FLamby. We rely on a common API for all strategies, which allows for efficient benchmarking of both datasets and strategies, as shown in Listing 1. As we focus on the cross-silo setting, we restrict ourselves to strategies with full client participation.

**FedAvg [99].** FedAvg is the simplest FL strategy. It performs iterative round-based training, each round consisting in local mini-batch updates on each client followed by parameter averaging on a central server. As a convention, we choose to count the number of local updates in batches and not in local epochs in order to match theoretical formulations of this algorithm; this choice also applies to strategies derived from FedAvg. This strategy is known to be sensitive to heterogeneity when the number of local updates grows [85, 77].

**FedProx [85].** In order to mitigate statistical heterogeneity, FedProx builds on FedAvg by introducing a regularization term to each local training loss, thereby controlling the deviation of the local models from the last global model.

**Scaffold [77].** Scaffold mitigates client drifts using control-variates and by adding a server-side learning rate. We implement a full-participation version of Scaffold that is optimized to reduce the number of bits communicated between the clients and the server.

**Cyclic Learning [22, 116].** Cyclic Learning performs local optimizations on each client in a sequential fashion, transferring the trained model to the next client when training finishes. Cyclic is a simple sequential baseline to other federated strategies. For Cyclic, we define a round as a full cycle throughout all clients. We implement both such cycles in a fixed order or in a shuffled order at each round.

**FedAdam [110]**, **FedYogi [110]**, **FedAdagrad [110].** FedAdam, FedYogi and FedAdagrad are generalizations of their respective single-centric optimizers (Adam [79], Yogi [134] and Adagrad [96]) to the FL setting. In all cases, the running means and variances of the updates are tracked at the server level.

```python
# Import relevant dataset, strategy, and utilities
from flamby.datasets.fed_camelyon16 import FedCamelyon16, Baseline, BaselineLoss, NUM_CLIENTS, metric
from flamby.strategies import FedProx
from flamby.utils import evaluate_model_on_tests, get_nb_max_rounds

# Define number of local updates and number of rounds
num_updates = 100
num_rounds = get_nb_max_rounds(num_updates)
# Dataloaders for train and test
training_dataloaderss = [
    DataLoader(FedCamelyon16(center=i, train=True, pooled=False), batch_size=BATCH_SIZE, shuffle=True)
    for i in range(NUM_CLIENTS)
]
test_dataloaders = [
    DataLoader(FedCamelyon16(center=i, train=False, pooled=False), batch_size=BATCH_SIZE, shuffle=False)
    for i in range(NUM_CLIENTS)
]
# Define local model and loss
model_baseline = Baseline()
loss_baseline = BaselineLoss()
# Define and train strategy
strategy = FedProx(training_dataloaders, model_baseline, loss_baseline, torch.optim.SGD, LR, num_updates, num_rounds)
model_final = strategy.run()[0]
# Evaluate final FL model on test sets
results_per_client = evaluate_model_on_tests(model_final, test_dataloaders, metric)
```

Listing 1: Code example from the FLamby dataset suite: on the Fed-Camelyon16 dataset, we use the FedProx Federated Learning strategy to train the pre-implemented baseline model.

## 3.4 Dataset Heterogeneity

We qualitatively illustrate the heterogeneity of the datasets of FLamby. For each dataset, we compute a relevant statistical distribution for each client, which differs due to the differences in tasks and modalities of the datasets. We comment the results displayed in Figure 1 in the following. Appendix M provides a more quantitative exploration of this heterogeneity.

For the **Fed-Camelyon16** dataset, we display the color histograms (RGB values) of the raw tissue patches in each client. We see that the RGB distributions of both clients strongly differ. For both **Fed-LIDC-IDRI** and **Fed-KITS2019** datasets, we display histograms of voxel intensities. In both cases, we do not note significant differences between clients. For the **Fed-IXI** dataset, we display the histograms of raw T1-MRI images, showing visible differences between clients. For **Fed-TCGA-BRCA**, we display Kaplan-Meier estimations of the survival curves [75] in each client. As detailed in Appendix F, pairwise log-rank tests demonstrate significant differences between some clients, but not all. For the **Fed-ISIC2019**, we use a 2-dimensional UMAP [98] plot of the features extracted from an Imagenet-pretrained Efficientnetv1 on the raw images. We see that some clients are isolated in distinct clusters, while others overlap, highlighting the heterogeneity of this dataset. Last, for the **Fed-Heart-Disease** dataset, we display histograms for a subset of features (age, resting blood pressure and maximum heart rate), showing that feature distributions vary between clients.

## 4 FL Benchmark Example with FLamby

In this section, we detail the guidelines we follow to perform a benchmark and provide results thereof. These guidelines might be used in the future to facilitate fair comparisons between potentially novel FL strategies and existing ones. However, we stress that FLamby also allows for any other experimental setup thanks to its modular structure, as we showcase in Appendices L.1 and L.2. The FLamby suite further provides a script to automatically reproduce this benchmark based on configuration files.

**Train/test split.** We use the per-client train/test splits, including all clients for training. Performance is evaluated on each local test dataset, and then averaged across the clients. We exclude model personalization from this benchmark: therefore, a single model is evaluated at the end of training. We refer to Appendix L.2 for more results with model personalization.

**Hyperparameter tuning and Baselines.** We distinguish two kinds of hyperparameters: those related to the machine learning (ML) part itself, and those related to the FL strategy. We tune these parameters separately, starting with the machine learning part. All experiments are repeated with 5 independent runs, except for FED-LIDC-IDRI where only 1 training is performed due to a long training time.

For each dataset, the ML hyperparameters include the model architecture, the loss and related hyperparameters, including local batch size. These ML hyperparameters are carefully tuned with cross-validation on the pooled training data. The resulting ML model gives rise to the **pooled baseline**. We use the same ML hyperparameters for training on each client individually, leading to **local baselines**.

For the FL strategies, hyperparameters include e.g. local learning rate, server learning rate, and other relevant quantities depending on the strategies. For each dataset and each FL strategy, we use the same model as in the pooled and local baselines, with fixed hyperparameters. We then only optimize FL strategies-related hyperparameters.

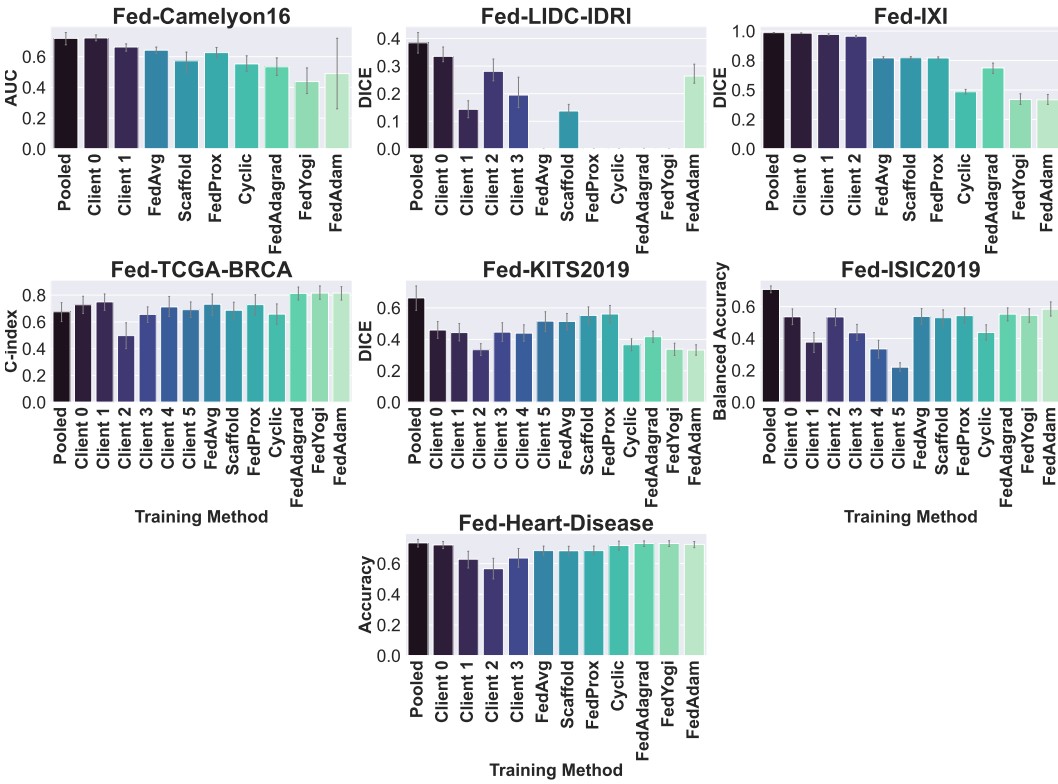

Figure 2: Benchmark results on FLamby for each dataset. For all metrics, higher is better, see Section 3.2 for metric details and Section 4 for experimental details. For Fed-LIDC-IDRI, multiple strategies fail converging, leading to zero DICE. Except for Fed-TCGA-BRCA and Fed-Heart-Disease, federated strategies fall short of reaching the pooled performance, but improve over the local ones.

**Federated setup.** For all strategies and datasets, the number of rounds $T_{\max}$ is fixed to perform approximately as many epochs on each client as is necessary to get good performance when data is pooled. Note that, as we use a single batch size $B$ and a fixed number $E$ of local steps, the notion of epoch is ill-defined; we approximate it as follows. Given $n_{epochs}^P$, the number of epochs required to train the baseline model for the pooled dataset, $n_T$ the total number of samples in the distributed training set, $K$ the number of clients, we define

$$T_{\max} = n_{epochs}^P \cdot \lfloor n_T/K/B/E \rfloor \tag{1}$$

where $\lfloor \cdot \rfloor$ denotes the floor operation. In our benchmark, we use $E = 100$ local updates for all datasets. Note that this restriction in the total number of rounds may have an impact on the convergence of federated strategies. We refer to Appendix J for more details on this benchmark.

**Benchmark results.** The test results of the benchmark are displayed in Figure 2. Note that test results are uniformly averaged over the different local clients. We observe strikingly different behaviours across datasets.

No local training or FL strategy is able to reach a performance on par with the pooled training, except for Fed-TCGA-BRCA and Fed-Heart-Disease. It is remarkable that both of them are tabular, low-dimensional datasets, with only linear models. Still, for Fed-KITS2019 and Fed-ISIC2019, some FL strategies outperform local training, showing the benefit of collaboration, but falling short of reaching pooled performance. For Fed-Camelyon16, Fed-LIDC-IDRI and Fed-IXI, the current results do not indicate any benefit in collaborative training.

Among FL strategies, we note that for the datasets where an FL strategy outperforms the pooled baselines, FedOpt variants (FedAdagrad, FedYogi and FedAdam) reach the best performance. Further, the Cyclic baseline systematically underperforms other strategies. Last, but not least, FedAvg does not reach top performance among FL strategies, except for Fed-Camelyon16 and Fed-IXI, it remains a competitive baseline strategy.

These results show the difficulty of tuning properly FL strategies, especially in the case of heterogeneous cross-silo datasets. This calls for the development of more robust FL strategies in this setting.

## 5 Conclusion

In this article we introduce FLamby, a modular dataset suite and benchmark, comprising multiple tasks and data modalities, and reflecting the heterogeneity of real-world healthcare cross-silo FL use cases. This comprehensive benchmark is needed to advance the understanding of cross-silo healthcare data collection on FL performance.

Currently, FLamby is limited to healthcare datasets. In the longer run and with the help of the FL community, it could be enriched with datasets from other application domains to better reflect the diversity of cross-silo FL applications, which is possible thanks to its modular design. Regarding machine learning backends, FLamby only provides PyTorch [104] code: supporting other backends, such as TensorFlow [4] or JAX [17], is a relevant future direction if there is such demand from the community. Further, our benchmark currently does not integrate all constraints of cross-silo FL, especially privacy aspects, which are important in this setting.

In terms of FL setting, the benchmark mainly focuses on the heterogeneity induced by natural splits. In order to make it more realistic, future developments might include in depth study of Differential Privacy (DP) training [42], cryptographic protocols such as Secure Aggregation [16], Personalized FL [43], or communication constraints [113] when applicable. As we showcase in Appendices L.1 for DP and L.2 for personalization, the structure of FLamby makes it possible to quickly tackle such questions. We hope that the scientific community will use FLamby for cross-silo research purposes on real data, and contribute to further develop it, making it a reference for this research topic.

## Acknowledgments and Disclosure of Funding

The authors thank the anonymous reviewers, ethics reviewer, and meta-reviewer for their feedback and ideas, which significantly improved the paper and the repository. The authors listed as Owkin, Inc. employees are supported by Owkin, Inc. The works of E.M. and S.A. is supported, in part, by gifts from Intel and Konica Minolta. This work was supported by the Swiss State Secretariat for Education, Research and Innovation (SERI) under contract number 22.00133, by the Inria Explorator Action FLAMED and by the French National Research Agency (grant ANR-20-CE23-0015, project PRIDE and ANR-20-THIA-0014 program `AI_PhD@Lille`). This project has also received funding from the European Union's Horizon 2020 research and innovation programme under grant agreement No 847581 and is co-funded by the Region Provence-Alpes-Côte d'Azur and IDEX UCAJEDI. A.D.'s research was supported by the *Statistics and Computation for AI* ANR Chair and by *Hi!Paris*. C.P. received support from *Accenture Labs* (Sophia Antipolis, France).

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
