# OpenReview forum: "FLamby: Datasets and Benchmarks for Cross-Silo Federated Learning in Realistic Healthcare Settings"
_NeurIPS.cc/2022/Track/Datasets_and_Benchmarks — NeurIPS 2022 Datasets and Benchmarks _

### Official Review · Reviewer_tsLT · 2022-07-24
**Good benchmark for cross-silo FL with a focus on medical computing tasks and some minor weaknesses**

**Rating:** 7
**Confidence:** 4

**Strengths:**

1. The size of the benchmark (7 datasets) is considerable. The datasets are also naturally partitioned.
2. The provided source code is relatively easy to follow, well documented, and has a high degree of completeness.
3. The predictive tasks are reasonably diverse (including classification, 3D segmentation, and a survival ranking problem).
4. The authors used various metrics to illustrate the statistical heterogeneity across clients (section 3.4)
5. Sufficient details (hyperparameters, training setup, code documentation) are included in the main text and the supplementary for reproducibility.
6. Literature review on past FL benchmarks is quite thorough.

Overall, the proposed benchmark/datasets are a valuable contribution to the FL community, though I encourage the authors to consider making it explicit that FLamby as it is right now mainly covers the cross-silo FL in the healthcare domain (see weaknesses).


**Weaknesses:**

1. While the title/abstract/introduction presents FLamby as an extensive benchmark for cross-silo FL, the datasets focus heavily on the healthcare domain (all 7 datasets) and image tasks (5 out of 7). I would suggest making the focus on healthcare more explicit in the title/abstract/introduction, as FLamby as it is right now is not representative of many other cross-silo settings, such as collaborations between banks, schools, and other institutions or data silos.
2. All datasets have a small number of clients (2 - 6), which may be specific to the healthcare domain. This may not be sufficient for benchmarking existing algorithms whose utility may benefit from a (slightly) larger client count. Similar to weakness #1, for a comprehensive cross-silo benchmark there should be at least one dataset with say 20-100 clients. It might be worth considering hybrid partitioning (i.e. having both artificial and natural partitions) and giving recommendations for practitioners on how to do that for the presented datasets.
3. Lack of personalization algorithms. The nature of cross-silo FL (particularly the properties that the clients are stateful, resource-abundant, and have relatively large local datasets) often necessitates model personalization. This is also suggested in Fig 3 where simple local training seems to be a strong baseline for many datasets.
4. (Minor) The presented tasks tend to require domain-specific knowledge (for data preprocessing, training, tuning, etc) and might make it slightly harder for practitioners to develop/adapt/tune FL algorithms. Compared to LEAF, the medical computing tasks and baseline models like Cox Model / nnU-Net / V-Net can be unfamiliar to many FL practitioners.

I would be happy to adjust my rating if the authors could address these concerns.

**Additional Feedback:**

- If the goal of the paper is to provide a comprehensive benchmark and evaluation, I think it is also acceptable (and encouraged) to incorporate existing (non-medical) cross-silo datasets into the benchmark, as standardizing the evaluation methods and providing recommendations/code would by itself be considered a good contribution. Otherwise, as per weakness #1, I would encourage the authors to make clear that FLamby is focused on the healthcare domain.
- L175: is there supposed to be a footnote (3)?
- L275: is the average client performance a weighted or a uniform average?
- L284 (Federated setup): if I understand correctly, the setup here (eq 1) basically says the number of FL rounds is determined by the time to train the “pooled” dataset, but (1) this might not be a good strategy since in practice the convergence of training on central, “pooled” dataset is often not the same as the convergence of FL training due to heterogeneity and (2) this heuristic might be a bit overcomplicated and one can simply train till convergence by setting a large enough $T_\text{max}$.

**Clarity:**

The paper is generally well-written and easy to follow.

Some minor issues:
- It may be worth simplifying the dataset names; e.g. “Fed-LIDC-IDRI” and “Fed-TCGA-BRCA” is a bit complicated, error-prone, and not very informative.
- Throughout the paper and the code, it seems that the term “center” or “local” is used to refer to “clients” in FL. For consistency please consider using the term “client” throughout.
- Fig 1(a) is not very easy to interpret; is the x-axis the pixel values? The goal seems to be to illustrate the heterogeneity across the two clients, but color histograms may not be the most informative figure, especially when multiple colors overlap. One possible alternative for better visualization is to juxtapose some form of “mean image” from each of the silos. If the authors insist on keeping Fig 1(a) please consider separating the colors.
- For Fig 1, consider changing “Local” in legends to “Client” for consistency and to avoid confusion.


**Correctness:**

- The presented datasets are constructed in a sound manner (e.g. authors emphasized having data naturally partitioned, and they performed good analysis on the resulting data heterogeneity).
- The metrics for evaluating the datasets look reasonable.
- The chosen baseline methods are reasonable, though the authors should consider personalization baselines as per weakness #3.
- The experiment setup is also reasonable.


**Documentation:**

Dataset documentation:
- The code provided in the supplementary seems to contain sufficient detail on how the datasets can be accessed and used. The datasets presented in this paper are available for public use but are hosted at different sources.

Benchmark documentation:
- The appendix and provided code seem to have sufficient detail to support reproducibility.


**Ethics:**

No ethical issues are identified.

**Relation To Prior Work:**

The literature review is fairly comprehensive and the paper discussed key distinctions from previous work.

**Summary And Contributions:**

- The paper presents 7 datasets with natural partitions for cross-silo FL settings. The presented datasets are from the healthcare domain, spanning multiple predictive tasks.
- The paper provides detailed descriptions and implementations of how to run standard FL algorithms on these datasets.
- The paper presents some analysis to illustrate the statistical heterogeneity of the datasets.
- The paper provides some initial benchmark results based on the implementation.

-----------------------

Post-rebuttal update: score raised from 6 to 7

---

> ### Author Response · Authors · 2022-08-19
> **Response to reviewer tsLT**
>
> We thank reviewer tsLT for their positive and thorough review. In addition to the general comment, we answer below all their comments. We believe these answers and the revised article address most of their comments, and we would politely ask the reviewer to revise their score based on them. We are open to further discussion with the reviewer regarding these points as well as the revised paper.
>
> ### Healthcare focus
> Healthcare is one of the domains where cross-silo FL has the biggest potential, which explains why we focused early on this domain, as our goal is to bridge the gap between research and applications. In order to better reflect this focus, we have changed the abstract and the introduction in the revised version. However, in the longer term and with the help of the FL community, the dataset suite could be easily extended to other domains.
>
> ### Number of clients
> As the reviewer points out, the number of clients is a consequence of the current healthcare focus. We thank the reviewer for the suggestion of hybrid partitioning. We added [reproducible scripts](https://github.com/owkin/FLamby/blob/main/flamby/datasets/split_utils.py) to provide such hybrid partitions on the datasets with an arbitrary number of clients, following a simple methodology explained in Appendix K.
>
> ### Lack of personalization algorithms
> Thanks for this comment. In order to showcase the possibility of investigating such approaches with FLamby, we have added a simple baseline of [Federated Averaging followed by local fine-tuning](https://github.com/owkin/FLamby/blob/main/flamby/strategies/fed_avg_fine_tuning.py) that we use to perform a simple experiment in Appendix K in the revised article.
>
> ### Domain-specific knowledge
> We would like to highlight that one of the advantages of FLamby is to make such tasks more accessible to FL researchers, thanks to standardized API. FLamby users can simply load the relevant dataset, model and loss functions. Although the reviewer is right in pointing out that some delicate work is still needed in tuning hyperparameters, FLamby considerably reduces the entry cost and hopefully will bring FL research closer to applications.
>
> ### Clarity suggestions
> Thanks for the suggestions. We corrected the issues with “local” and “clients”. We will ensure to take into account all of your remarks for the camera-ready version.
>
> ### Client performance average
> Clients’ performances are uniformly averaged. We explicited it in the revised version of the article.
>
> ### Number of rounds to reach convergence
> Thanks for the comment. We would like to highlight that the FLamby suite allows for arbitrary modifications in the experimental setup, including the ones suggested by the reviewer; what is most important is the pooled baseline performance that will be used for comparison to FL strategies’ performance.

---

### Official Review · Reviewer_1pXz · 2022-07-26
**Good cross-silo medical FL datasets**

**Rating:** 6
**Confidence:** 3
**Clarity:** The paper is written quite well and w…

**Strengths:**

- The paper provides 7 different datasets in the cross-silo FL settings, across different training tasks from binary/multi-class classification, 3D segmentation and survival.
- The papers also provide quite extensive FL benchmarks across different strategies, such as FedAvg, FedProx, SCAFFOLD, FedAdam and Cyclic learning, with details experimental setups and hyper-parameters.
- The paper provides details for each datasets with statistics and heterogeneity studies.

**Weaknesses:**

- The paper mentioned that the cross-silo situation is normally for small client number, when client numbers are from 2 to 50, with each clients having medium to large data size. However, the maximum number of clients among these 7 datasets are 6, which is nowhere near 50. As for the size of dataset, the Fed-KITS2019 only has 96 data samples in total, with some clients only has 12 sample.
- As the paper also mentioned, the datasets are limited to healthcare datasets, which can be limited in terms of cross-silo FL applications. As paper mentioned that there are other stand-alone cross-silo FL datasets, which are not healthcare dataset.

**Additional Feedback:**

A few points that need to clarify:

- The paper mentioned that apart form Fed-KITS2019, the pooled performance is not attained by FL strategies. However, by looking at Fed-Camelyon16 in Fig 3, it seems both Cyclic Learning and FedAdagrad are better than pooled?
- (Minor point) Section 3.1, authors mentioned that FLamby can be compatible with existing FL libraries, just checking have authors implement FLamby datasets in the mentioned FL libraries?

**Correctness:**

- On line 286, the paper said E stands for number of local steps. Does it means number of SGD steps or number of local epochs? It also mentioned that you use E=100 local updates for all datasets, which means that number of local updates are fixed regardless of the number of data samples in each clients. Normally, FL reports in terms of number of local epoch and number of communication rounds. If it means number of local epoch, 100 seems quite a lot, given some of clients only have 12 samples. If it means number of SGD steps, it seems not reasonable to use the same number of SGD steps for different number of data samples.

**Documentation:**

All 7 datasets are not new datasets, but rather collected from existing datasets are partitioned by the hospitals/institutes to provide real scenarios in the FL settings.

**Relation To Prior Work:**

It provides the benchmarks for cross-silo FL scenarios, which as the paper stated, are less easy to find in the current literature. There is LEAF benchmarks that are primarily focus on cross-devices FL settings.

**Summary And Contributions:**

The paper focuses on the cross-silo federated learning setting, where the number of clients are usually small but the data size per clients are usually medium to large. The typical clients for the cross-silo settings can be hospitals, banks or various institutes. The paper doesn't provide new datasets, but rather use the existing dataset and partitioned the datasets by the data collecting hospitals/institutes. The datasets consist of different task type: classification, segmentation or survival. The paper also open-sources the code and provides benchmarks for different FL strategies, including FedAvg, FedProx, SCAFFOLD, FedAdam, and Cyclic learning.

---

> ### Author Response · Authors · 2022-08-19
> **Response to reviewer 1pXz**
>
> We thank reviewer 1pXz for their positive and thorough review. In addition to the general comment, we answer below all their comments. We are open to further discussion with the reviewer regarding these points as well as the revised paper.
>
> ### Number of silos and dataset size
> Regarding dataset size, we note that the datasets vary in sizes, ranging 96 to 23k. The relative small size of these datasets compared to other fields is due to the current focus in healthcare (cf below). Further, we would like to point out that having silos with extremely small cohorts (12-15) can happen e.g. in rare diseases, where FL could play a huge role; as such, this makes this dataset realistic and close to applications.
>
> Regarding the number of silos, we have updated the codebase to open the possibility for more clients by synthetically splitting existing silos in a reproducible fashion (see Appendix K).
>
> ### Healthcare focus
> Healthcare is one of the domains where cross-silo FL has the biggest potential, which explains why we focused early on this domain, as our goal is to bridge the gap between research and applications. In order to better reflect this focus, we have changed the abstract and the introduction in the revised version. However, in the longer term and with the help of the FL community, the dataset suite could be easily extended to other domains.
>
> ### Number of local steps
> Thanks for this comment. $E$ denotes the number of SGD steps, and not epochs. We respectfully disagree with the remark of the reviewer regarding the need for performing epochs-based updates. As explained in FedNova [1], this approach leads to biases and prevents from optimizing the correct loss function, while having the same amount of local SGD steps does not suffer from this issue.
>
> [1] Wang et al. Tackling the Objective Inconsistency Problem in Heterogeneous Federated Optimization, NeurIPS 2020
>
> ### Results comments
> Thanks for this remark.There were indeed some inconsistencies in the discussion of the results. They have been corrected in the revised version of the article, see lines 310 to 324 in the revised version.
>
> ### Compatibility with external libraries
> We have provided examples of integration of FLamby with [FedML](https://github.com/owkin/FLamby/blob/main/integration/FedML/flamby-integration-into-fedml.md) and [Fed-BioMed](https://github.com/owkin/FLamby/blob/main/integration/fedbiomed/flamby-integration-into-fedbiomed.md) in the now open source repository.

---

> ### Author Response · Authors · 2022-08-26
> **Response to Reviewer 1pXz**
>
> We thank the reviewer again for their feedback and hope we have addressed their questions/concerns. If there are any additional questions, we would be happy to answer them.

---

### Official Review · Reviewer_KtyC · 2022-07-26
**A dataset suite and benchmark for cross-silo FL**

**Rating:** 5
**Confidence:** 4
**Correctness:** I believe the methodology used for da…
**Clarity:** It's a clear paper, but I would expec…

**Strengths:**

They provide standardized dataset APIs for 7 realistic healthcare datasets.

These dataset APIs are agnostic to existing FL libraries for easy and flexible evaluation.

It's easy to conduct the benchmark with a few lines of code.

**Weaknesses:**

1. These 7 datasets are relatively small compared to the cross-device FL datasets. Additionally, limited to the healthcare domain as well as the tasks.

2. No privacy, security, and fairness strategies were introduced and benchmarked.

3. The baseline FL strategies are directly from cross-device FL. What about cross-silo FL algorithms?

4. Limited contribution in the benchmark. If all these are simulations inside a cluster, what are the key differences between your benchmark and with existing cross-device FL benchmark?

Even the authors mentioned some points above in future work, but the current version has very limited contribution to the cross-silo FL.

**Additional Feedback:**

Please refer to the sections above.

**Documentation:**

The dataset description is in great detail in this paper.

**Relation To Prior Work:**

As I discussed in the weakness sections.

**Summary And Contributions:**

This paper proposes an open-source cross-silo federated dataset and benchmark called FLamby.

Summarized contributions
1. Build an open-source federated cross-silo dataset suite including 7 datasets across different tasks in multiple domains in healthcare.

2. Provide guidelines to benchmark FL strategies

3. Make it accessible for benchmark reproducibility and easy integration in different FL frameworks

---

> ### Author Response · Authors · 2022-08-19
> **Response to reviewer KtyC**
>
> We thank reviewer KtyC for their review. In addition to the general comment, we provide a detailed answer to all their comments below. Based on these answers, we would like to politely ask the reviewer to re-evaluate the revised paper. We are open to further discussion with the reviewer regarding the revised paper.
>
> > 1. These 7 datasets are relatively small compared to the cross-device FL datasets. Additionally, limited to the healthcare domain as well as the tasks.
>
> We tackle this comment below in 3 parts:
>
> **1.1 “Limited to the healthcare domain”**
>
> Healthcare is one of the domains where cross-silo FL has the biggest potential, which explains why we focused early on this domain, as our goal is to bridge the gap between research and applications. In order to better reflect this focus, we have changed the abstract and the introduction in the revised version. However, in the longer term and with the help of the FL community, the dataset suite could be easily extended to other domains.
>
> **1.2 These 7 datasets are relatively small compared to the cross-device FL datasets**
>
> Thank you for this comment. This is a consequence of the current healthcare focus. Indeed, medical datasets are characterized by a paucity of data compared to other fields. As we show in this work, this aspect may critically affect the performance of state-of-the-art FL strategies, and motivates further research on this topic. We hope that FLamby will help researchers devise FL strategies which are applicable in this setting, thereby bridging the gap between FL research and applications. In the longer run, more massive datasets from other domains might also be included.
>
> **1.3 “Limited to the healthcare domain *as well as the tasks”***
>
> We understand the reviewer means that the tasks covered by the benchmark are healthcare-specific. We respectfully disagree with this comment. As displayed in Figure 1, the benchmark covers 3 task types: classification (binary or multiclass), 3D segmentation and survival analysis. Although the datasets they are applied to stem from healthcare, none of the tasks is specific to healthcare.
> Classification is a transversal task in most of today’s machine learning;
>
> 3D segmentation is a “fundamental and challenge problem in computer vision and graphics” [1], with applications in autonomous driving, mobile robots, industrial control, augmented reality and medical image analysis [1];
>
> Survival analysis is not only used in biostatistics, but is also a fundamental tool in actuarial studies [2] as well as in predictive maintenance [3].
>
> As a consequence, the presented tasks would not be of interest solely for healthcare researchers, but also potentially for the insurance and the industry, where cross-silo FL applications also make sense.
>
>
> [1] He et al, Deep Learning based 3D segmentation: A Survey, ArxiV preprint 2021 under review of ACM Computing Surveys
> [2] Sari et al., Pricing life insurance premiums using Cox regression model, AIP Conference Proceedings, 2019
> [3] Chen et al., Predictive maintenance using Cox proportional Hazard Deep learning, Advanced Engineering Informatics, 2020
>
> > 2. No privacy, security, and fairness strategies were introduced and benchmarked.
>
> The goal of FLamby is to bridge the gap between research and applications, by bringing realistic cross-silo medical FL datasets in the hands of researchers. In the benchmark, we focused on simplicity, modularity, and lightness, to facilitate adoption. In this view, adding privacy or fairness features would not bring much value. We further note that standard cross-device FL benchmarks, such as LEAF, do not incorporate such features. However, thanks to the extensivity of the benchmark, it is easy for researchers to investigate these important topics with FLamby. This can be done either on top of FLamby’s codebase, as we showcase with a differentially private training example (Appendix L.1 in revised version), or via external FL libraries with such features, as FedML and Fed-BioMed, for which we show [examples of integration](https://github.com/owkin/FLamby/tree/main/integration).
>
> > 3. The baseline FL strategies are directly from cross-device FL. What about cross-silo FL strategies?
>
> We respectfully disagree with this comment. We included Scaffold [1] in the benchmark, which is a cross-silo FL strategy as it assumes that the clients are stateful.
>
> [1] Karimireddy et al., Scaffold: Stochastic Controlled Averaging for Federated Learning, ICML 2020

---

> ### Author Response · Authors · 2022-08-19
> **Response to reviewer KtyC Part 2**
>
> > 4. If all these are simulations inside a cluster, what are the key differences between your benchmark and with existing cross-device FL benchmark?
>
> FLamby is a cross-silo FL benchmark, which does not make the same assumptions as cross-device FL. Further, we would like to point out that the codebase was designed to be simple and lightweight, as most of FL research takes place in simulations. In order to perform experiments on different machines, we highlight the compatibility with other frameworks, such as FedML and Fed-BioMed, which opens this possibility. In particular, [the integration with Fed-BioMed](https://github.com/owkin/FLamby/blob/main/integration/fedbiomed/flamby-integration-into-fedbiomed.md) runs on docker containers.

---

> ### Author Response · Authors · 2022-08-26
> **Response to Reviewer KtyC**
>
> We thank the reviewer again for their feedback and hope we have addressed their questions/concerns. If there are any additional questions, we would be happy to answer them.

---

### Official Review · Reviewer_NALj · 2022-07-27
**Review for FLamby**

**Rating:** 7
**Confidence:** 4

**Strengths:**

* The paper does a very good job at organising datasets from the medical domain in the cross-silo setting, federating them, and providing a benchmark implementation of a model that works. This can help a lot of people bootstrap by having a set of baselines along with dataloaders they can extend upon.
* The authors do a very good work at qualitatively explaining the setup and task of each dataset, along with the data heterogeneity that describes each one.
* I appreciate the effort that the authors have put in developing out-of-the-box solutions for each task at hand, covering a wide range of tasks and modalities.

**Weaknesses:**

* Flamby is not introducing a dataset per se, but rather organises and adapts existing datasets in the realm of cross-silo federated learning to democratise participation by offering a set of baselines.
* The datasets integrated in the Flamby are focusing rather on the smaller side of cross-silo, particularly in terms of number of participating clients (<10). Moreover, the case of vertically split cross-silo FL has not been visited at all. This hurts a bit the generality and scalability of the proposed suite.
* The paper, in its current state, is not a suite for general cross-device FL, but rather more focused on the medical domain.

**Additional Feedback:**

I honestly like this paper and believe it is moving the needle towards a direction if FL worth pursuing. My main concern is that it appears to be more general than it is.

I am also documenting here some missing things that I would like to have seen in the paper:

* Introduction: I would also like to have seen the issue of system heterogeneity as a trait of cross-device vs. cross-silo setups (i.e. discrepancy in the capabilities of participating clients). Moreover, the general availability and reliability of these clients is limited.
* lines 52-53, "[...] the other highlighted [...] synthetic partitioning": While intuitively true, I would expect a citation for this claim.
* On the simulated run setup:
    * Can workers run on different machines through the simulation script? If so, what is the communication protocol. If not, do different clients run sequentially or in multi-treaded/process setup?
    * How is the dataset distributed to different workers?
    * Is GPU acceleration supported? (missing from the paper)
* On experimental setup:
    * How many different runs have been run?
    * Experimental setup and time to convegence missing for assessing the approximate time cost of training.
    * It would be nice to have seen some convergence graphs in the appendix.

I don't think the requirement on the client-side for having a google account to download from google drive is necessary. The authors should adopt such an approach (in place of the GDrive's Python SDK) for open access.

Last, I think that having integrated a DP/SecAgg-bootstrapped simulation would have made the paper much more appealing. Treating privacy - especially in the realm of medical data - as orthogonal/future work can potentially lead to harmful side-effects. Maybe a comment could go to the "broader impact" section of the paper about this.

**Clarity:**

The paper is very well written with details about the setup, evaluation and reproducibility of the experiments.

Some minor suggestions include:
* Figures 1,3 have quite small font size.
* The algorithm in Figure 2 is not really a "figure". Maybe a different caption heading would be better. It might as well be moved to the appendix in favour of freeing up some space for results of Figure 3.
* Hyperparameter tuning and Baselines: This section could be written more clearly.
* Appendix, L9, "porigin": typo

**Correctness:**

* The paper, in its current state, is not a suite for general cross-device FL, but rather more focused on the medical domain. Albeit the authors' desire to become a more general suite, the papers fails to convince otherwise. This is actually one of the main points that I am putting the current score.
    * The authors talk in their related work about "extensive benchmarks with natural splits" being available in the cross-silo setting, referring to vision and HAR use-cases, but do not integrate any of these in Flamby.
    * Maybe it would be better to retitle the paper as "medical cross-silo" in its current form.
* Lines 121-122, "compatible with existing FL software libraries [...]": I am not sure where this compatibility stems from. Is it the fact that one can use the provided dataloaders? The provided simulation script is inherently incompatible with the workflow of many FL libraries and is to be used in place of these, not agnostically on top of these.
    * The authors focus their effort in a PyTorch compatible implementation. Is there an intention of extending it to other framework backends (TFF, etc.)?
* In terms of the local adaptive optimisers, it is unclear how the authors aggregate these.

**Documentation:**

Documentation is sufficient. If the authors want to go the extra mile, they could offer a container (e.g. docker) with everything set up when they open up their repository.

**Ethics:**

The authors have sufficiently covered this. If anything, I would urge the authors to further highlight the need for privacy-preserving techniques to be applied on top of cross-silo FL.

**Relation To Prior Work:**

It has been largely the case that many frameworks provide pre-implemented dataloaders for FL datasets (e.g. FedScale, TFF, FedML, FLSim), strategies, models, etc. to enable users bootstrap their research/development.

I see the proposed approach as an attempt to standardise experiments/datasets through benchmarking suites, that are subsequently being inherited by FL frameworks and provided off-the-shelf by the respective tool.

In the related work, it would be conducive to the reader to see what cross-silo datasets major FL frameworks provide and potentially how Flamby's implementation differs if there is an intersection. This could also be relevant to FL framework developers to fix potential issues in their current implementations or provide additional support for extra datasets.

**Summary And Contributions:**

This paper is introducing a dataset benchmark suite for cross-silo FL in medical applications. In particular, it gathers 7 previously available datasets, which the author federate, and provides benchmark models and FL glue code (simulated setup) for running experiments on "realistic" cross-silo setups. Coarsely put, one can think of it as LEAF for cross-silo Federated Learning.

---

> ### Author Response · Authors · 2022-08-19
> **Response to reviewer NALj 1/2**
>
> We thank reviewer NALj for their thorough review. In addition to the general comment, we provide a detailed answer to all their comments below. Based on these answers, we would like to politely ask the reviewer to re-evaluate the revised paper. We are open to further discussion with the reviewer regarding the revised paper.
>
> ### Medical focus
>
> Healthcare is one of the domains where cross-silo FL has the biggest potential, which explains why we focused early on this domain, as our goal is to bridge the gap between research and applications. In order to better reflect this focus, we have changed the title, abstract, and introduction in the revised version. However, in the longer term and with the help of the FL community, the dataset suite could be easily extended to other domains.
>
> ### Dataset size
>
> We would like to point out that we provide 7 datasets with number of data points varying from 96 to 23k+. The relative small size of the datasets compared to other fields, e.g. computer vision, stems from the medical nature of the datasets, which are notably characterized by small cohort sizes. Regarding the number of silos, we have updated the codebase to allow for the possibility to increase the number of clients (artificially, by splitting each natural client) in a reproducible fashion, see Appendix K in the revised version.
>
> ### Related cross-silo FL works
> > The authors talk in their related work about "extensive benchmarks with natural splits" being available in the cross-silo setting, referring to vision and HAR use-cases, but do not integrate any of these in Flamby.
>
> We believe there is a misunderstanding. The reviewer refers to L99-100 of the original paper (L104-105 in revised version), where the full sentence reads “to the best of our knowledge, **no** extensive benchmark with natural splits is available for cross-silo FL” (we emphasised the “no” here). Indeed, one of the motivations of this work is to address this lack of cross-silo FL benchmarks, compared to the multiple cross-device FL benchmarks (e.g. LEAF).
>
> ### Compatibility with existing FL software libraries
> The compatibility is a consequence of the standardized low-level dataset API of the data loaders, baseline models and losses. In the now [open source code](www.github.com/owkin/flamby), we show examples of such [integrations with FedML and Fed-BioMed](https://github.com/owkin/FLamby/tree/main/integration).
>
> ### Deep learning backend
> We initially focused on pytorch as it is the most widely used framework in machine learning research [1, 2]. However, we will extend the benchmark to other backends, e.g. TensorFlow or JAX, if there is demand from the community. We have added this as a future direction in the revised paper.
>
> [1]https://www.assemblyai.com/blog/pytorch-vs-tensorflow-in-2022/
>
> [2]https://thegradient.pub/state-of-ml-frameworks-2019-pytorch-dominates-research-tensorflow-dominates-industry/
>
> ### Aggregation of client-level adaptive optimizers
>
> In the FL strategies we benchmark, only standard SGD updates are performed at the client-level, see Tables 11, 12, and 13 in the supplementary material. In particular, for the FedOpt strategies (FedAdam, FedYogi, FedAdagrad), the adaptative updates take place at the server level.
>
> ### Clarity
>
> Thanks for the useful feedback. We did our best to improve the figures in the revised version and will continue these efforts for the camera-ready version. In the revised version, Figure 2 has been renamed, the hyperparameter tuning and baselines section has been rewritten, and we have also corrected the typo you pointed out.
>
> ### Related work and FL frameworks
>
> Thank you for this useful remark. Our related work section already covered tensorflow-federated (L95 in original version, L98 in revised v.). We investigated the current versions of FedScale, FedML and FLSim pointed out by the reviewers, as well as FATE. None of them provides comprehensive cross-silo FL benchmarks with natural splits: most of the datasets they provide stem from previously introduced cross-device benchmarks, such as LEAF, or are split in a synthetic fashion across silos based on e.g. Dirichlet sampling with class labels. We have updated the related work accordingly in the revised version L100-102.
>
> ### Documentation / Docker container
>
> Thanks for the remark. We have added a dockerfile example in the repository with FLamby set up and an [example running on it with Fed-Heart-Disease](https://github.com/owkin/FLamby/tree/main/dockers).
>
> ### System heterogeneity
> Thanks for pointing this out. We have updated the paragraph related to cross-device FL in the revised version to incorporate this challenge (L35 in revised version).

---

> ### Author Response · Authors · 2022-08-19
> **Response to reviewer NALj 2/2**
>
> This response follows the 1st part.
>
> ### Synthetic partitioning and sources of heterogeneity
>
> Thanks for this comment. In the example of digital pathology L52-53, known factors of variability include demographics, staining techniques, storage methodologies and digitization process [1]. In Figure 1 of [1], one sees that staining normalization techniques exist, but other factors of variability are hardly addressed other than through multicentric cohort collection, as they are not completely understood.
>
> Indeed, zooming on the digitization process, it is known that “Scanner manufacturers may employ different approaches for slide digitization, including different hardware (eg, bulbs for lighting, charge-coupled device chips for digitization), algorithms for image manipulation (eg, stitching, compression), and file formats.” [2].  In turn, the effect of digitization is known to play a role in machine learning applications, see e.g. [3] where it is shown that the image compression level can lead to classification errors.
>
> Reproducing all the aforementioned effects in a synthetic fashion is a challenging task in itself. Further, we stress that these are *known* ones; there might be unknown ones which exist in multicentric cohorts but would be impossible to reproduce with synthetic data generation. This calls for multicentric cohort collection.
>
> We added these references and amended the text to better reflect this argument in the revised article, L50-55.
>
> [1] Howard et al., The impact of site-specific digital histology signatures on deep learning model accuracy and bias, Nature Communications, 2021, https://www.ncbi.nlm.nih.gov/pmc/articles/PMC8292530/
> [2] Janowczyk et al., HistoQC: An Open-Source Quality Control Tool for Digital Pathology Slides, JCO Clinical Cancer Informatics, 2019, https://www.ncbi.nlm.nih.gov/pmc/articles/PMC6552675/
> [3] Fu et al., Pan-cancer computational histopathology reveals mutations, tumor composition and prognosis, Nature Cancer, 2020 https://www.nature.com/articles/s43018-020-0085-8
>
> ### Simulated run setup
>
> FLamby was designed to be lightweight and simple code to enable ease of use and subsequent adoption. All clients run sequentially, without any multithreading. Datasets are simply assigned to clients as different python objects. GPU acceleration is supported thanks to the pytorch backend. We have added more precision on the setup in the revised appendix J.2 as well as in the code description in the repository.
>
> ### Experimental run setup
> Thank you for these useful remarks.
>
> Regarding the number of runs, we performed 5 independent runs with different random seeds, except for the largest one (Fed-LIDC-IDRI) where we only performed a single run due to runtime constraints. We added this piece of information in Appendix J.4 in the revised version.
>
> In the description of each dataset in the revised appendix, we added an estimate of the time to convergence of a single training. These times range from a minute for the lighter ones to two days for Fed-LIDC-IDRI.
>
> ### Google Drive API for dataset download
> Thanks for the feedback. We would like to point out that remark only applies to 1 out of 7 datasets (Fed-Camelyon16). This dataset is stored on a google drive and has nested folders, making the google drive API most practical to download these large files. Unfortunately, the google drive API requires to have a Google account. Note that for users without google account, an alternative option is to manually download the dataset from the browser interface. We updated the corresponding [instructions](https://github.com/owkin/FLamby/tree/main/flamby/datasets/fed_camelyon16) accordingly.
>
> ### Privacy simulations
>
> Thanks for this feedback. We agree with the reviewer that cross-silo FL is a complex topic with many intricate challenges, including privacy or personalization, which are not orthogonal to one another. We would like to point out that we added an example of differentially private training in the repository, showcasing the usage of FLamby to tackle these questions, as well as in Appendix L.1. We updated the conclusion to urge the community to tackle these questions on real data thanks to Flamby, and, following the reviewer’s suggestion, updated the Broader impact section.

---

> ### Author Response · Authors · 2022-08-26
> **Response to Reviewer NALj**
>
> We thank the reviewer again for their feedback and answer to our rebuttal and hope we have addressed all their questions/concerns. If there are any additional questions, we would be happy to answer them.

---

### Official Review · Reviewer_hEKK · 2022-07-27
**Review of "FLamby: Datasets and Benchmarks for Cross-Silo Federated Learning in Realistic Settings"**

**Rating:** 8
**Confidence:** 3
**Clarity:** The paper is clearly written.

**Strengths:**

- The datasets are easily accessible;
- The choices made in model selection, pre-processing, hyperparameter optimization, appear to be adapted and adequate for each dataset in the suite;
- The benchmark can be replicated and can also be extended with other datasets, models and/or FL strategies, without much development effort;
- The limitations of previous related work are well framed and are partially solved with the presented suite of datasets.

**Weaknesses:**

- The presented datasets only encompass medical tasks (e.g, cancer detection);
- The datasets have limited size;
- Most of the datasets have image data types;
- Although no new dataset is presented (the datasets are repurposed from previous uses), there is no discussion on the ethical/social implications nor documentation about the datasets, following one of the frameworks presented in the call.

**Additional Feedback:**

- The placement of images and tables could be improved. If they could be placed inline with the text instead of the top of the page, it would have a better flow with the text.
- Table 1 has cut text, due to the table size.
- It is not clear why the plots 1b, 1c, and 1d have different formats.
- The code snippet is interesting but it would make more sense to include it in supplementary material.
- The discussion on results and their implications (e.g., answering questions such as, "are FL strategies dataset-dependent?", "Which strategy is the better baseline?", etc.) could be extended.
- Having datasheets (or at least more extended documentation for each dataset, not focusing on the setup).

These are merely suggestions for improvement from my perspective in the paper. I encourage the authors to comment their thoughts on these points.



**Correctness:**

Yes, the claims of the submission are correct. The datasets are representative of one of the possible applications of FL, and they represent _cross-silo_ sources of data. The datasets appear to be collected and pre-processed in a sound way.

The benchmark performed is not exhaustive (only simpler FL strategies are employed), but the models, hyperparameter optimization, metrics, and FL strategies appear to be chosen appropriately accordingly to the dataset.



**Documentation:**

The documentation from this work is mainly focused on the reproducibility of the results. It focuses on describing the task, the download and pre-processing steps on the data, the baseline models and hyperparameter search parameters. There are no considerations on the ethical and responsible use of the data.

The datasets are accessible through an API, included in the code. The datasets are pulled from the original sources, and, therefore depend on third parties for revisions, versioning, etc. I recommend the authors to centralize the datasets that constitute the benchmark in a single source. This way, there is no risk in losing a dataset, or suddenly being incompatible with the code in the API.

**Ethics:**

The datasets included in the benchmark are from medical origin, and align with point 4 of the General Ethical Conduct (studies predicting characteristics (e.g., health status) from human data). Because of this, I believe the authors should include more information regarding the original datasets. It would be great to include the original datasheets if they exist or create new ones if they do not.

Additionally, a mechanism to revoke/update a dataset from the suite should exist.

**Relation To Prior Work:**

Yes, the work distinguishes itself from previous contributions. The main difference between this and previous works is the setting of FL (_cross-silo_), and the presentation of a benchmark.

**Summary And Contributions:**

This work introduces an aggregation of medical datasets which are relevant for the context of Federated Learning (FL). This means that each dataset is constituted by data gathered from different sources (referred as "clients"), with the same classification task and similar input.

These datasets differ from previously presented dataset suites for benchmarking FL, where the datasets are _cross-device_, where there are many clients, but each client provides a low number of instances. Here, the datasets are _cross-silo_, where the number of clients is reduced, but each client contributes with a large number of instances.

Additionally, the authors provide a benchmark of different FL algorithms, against the baselines of single client training and pooled training.

The code for downloading, splitting and replicating the benchmark results are made available in a python package.

---

> ### Author Response · Authors · 2022-08-19
> **Response to Reviewer hEKK**
>
> We thank the reviewer for the positive and insightful review. In addition to the general comment, we answer below all of their comments. We are open to further discussion with the reviewer regarding the revised paper.
>
> ### Medical focus
>
> Healthcare is one of the domains where cross-silo FL has the biggest potential, which explains why we focused early on this domain, as our goal is to bridge the gap between research and applications. In order to better reflect this focus, we have changed the title, abstract, and introduction in the revised version. However, in the longer term and with the help of the FL community, the dataset suite could be easily extended to other domains.
>
> ### Limited size
>
> The datasets’ sizes range from 96 to 23k. Although this is small compared to the scales typically found in computer vision, these sizes are standard in healthcare; as such, the benchmark reflects well the reality of applications.
>
> ### Image data types
>
> Out of the 7 datasets currently in FLamby, 5 are indeed types of images, and 2 of them tabular. However, we would like to point out that among images, we provide different types of images: histopathology slides ($10^5 \times 10^5$ pixels), 3D CT scans, as well as dermoscopy images, each featuring different technical challenges.
>
> ### Ethics and dataset document
>
> We thank the reviewer for this insightful remark which helped improve the submission. We have added an ethics section in the description of each dataset, both in the revised supplementary and in the online repository (individual readmes). We further added pointers to existing datasheets in the online repository.
>
> In the revised article, we also indicate that we will follow a maintenance plan, including measures to adapt the dataset suite to patients opting out from datasets or to update the dataset if need be (Appendix B.2).
>
> Regarding dataset storage, the main reason we limit data hosting is due to licenses conflicts; in particular, several datasets are released under creative commons licenses with share alike clauses, while FLamby is released under MIT license.
>
> ### Additional feedback
>
> Thanks for the useful feedback which helped us improve the article. We have taken most of it into account in the revised version and will take it fully into account in the camera-ready version.

---

> > ### Comment · Reviewer_hEKK · 2022-09-02
> > **Follow up response**
> >
> > I thank the authors for the time spent addressing the issues raised in my comment.
> >
> > There are significant improvements to the submission when compared to the original one. I appreciate the code quality and commitment to maintaining the package. I believe the final step in this cycle is to index the package in pypi and conda.
> >
> > I will update my grade accordingly.
> > Regards.

---

> ### Author Response · Authors · 2022-08-26
> **Response to Reviewer hEKK**
>
> We thank the reviewer again for their feedback and hope we have addressed their questions/concerns. If there are any additional questions, we would be happy to answer them.

---

### Official Review · Reviewer_JW67 · 2022-07-28
**An useful benchmark for cross-silo FL**

**Rating:** 4
**Confidence:** 3
**Clarity:** The paper is well written

**Strengths:**

1. Very useful datasets in healthcare domains
2. Partitioned by natural client information
3. Timely topic and relatively unexplored direction

**Weaknesses:**

1. The authors claim the benchmark to be comprehensive, but the scope is only limited to the healthcare domain.
2. The main contribution of the paper is introducing new datasets for cross-silo federated learning. However, FLamby's datasets are already available and not collected by the authors.
3. I think the paper should provide more details on the benchmark results and explain them. It will be very interesting to see if FLamby will enable the developers to get new insight into a cross-silo setting.
4. Security and privacy are major concerns in the cross-silo settings. (although mentioned in future work) FLambly should enable the developers to explore, for example, different cryptographic algorithms.
5. The scale of the dataset is relatively small, even for a cross-silo setting.


**Additional Feedback:**

Thanks for submitting to NeurIPS 2022 Datasets and Benchmarks Track. I believe this is an important topic and the proposed datasets will be very useful. However, I believe the paper can be improved by addressing the concerns I listed in the weakness section.

**Correctness:**

The claims are mostly correct. However, the paper lacks details on the evaluation metrics of the benchmark.

**Documentation:**

The paper is well-documented. However, the paper is not hosted online now.

**Ethics:**

There are no ethical concerns.

**Relation To Prior Work:**

The paper clearly discusses previous works and how it differs from them.

**Summary And Contributions:**

Fedamby is an open source cross-silo federated learning dataset with natural partitions, code examples, and benchmarking guidelines. It contains 7 datasets in the healthcare domain with different scales.

---

> ### Author Response · Authors · 2022-08-19
> **Response to reviewer JW67**
>
> We thank reviewer JW67 for their review and comments. We answer in the general comment as well as below all their comments. Based on these answers, we would like to politely ask the reviewer to re-evaluate the revised paper. We are open to further discussion with the reviewer regarding the revised paper.
>
> ### 1. Focus on healthcare
>
> Healthcare is one of the domains where cross-silo FL has the biggest potential, which explains why we focused early on this domain, as our goal is to bridge the gap between research and applications. In order to better reflect this focus, we have changed the title, abstract, and introduction in the revised version. However, in the longer term and with the help of the FL community, the dataset suite could be easily extended to other domains.
>
> ### 2. The main contribution of the paper is introducing new datasets for cross-silo federated learning. However, FLamby’s datasets are already available and not collected by the authors.
>
> - As we show in our related works section, to date, there is no curated collection of public datasets for cross-silo FL applications. Currently, there is a high heterogeneity on how researchers identify and use the datasets for FL applications, and no available standard. This affects reproducibility in this field.
>
> - Our contribution regarding these datasets consists in:
>   - Documenting these datasets;
>   - If necessary, finding metadata related to the centers to provide a natural split;
>   - Preparing download, preprocessing scripts, dataloading routines;
>   - Implementing client-level baselines (models, losses) reaching good results.
> - All of this is done in a unified and simple API. This makes it very easy for new researchers to re-use these datasets and focus on the federated learning algorithms. Ultimately, this serves our goal of bridging the gap between FL research community and practice.
>
> ### 3.  I think the paper should provide more details on the benchmark results and explain them. It will be very interesting to see if FLamby will enable the developers to get new insight into a cross-silo setting.
>
> Thank you for this comment. We expanded the benchmark results section in the revised version. Enabling practitioners to gather new insights on the cross-silo FL setting is one of the primary goals of this work.
>
> ### 4. Security and privacy are major concerns in the cross-silo settings. (although mentioned in future work) FLambly should enable the developers to explore, for example, different cryptographic algorithms.
>
> We focused on lightness and ease of use to enable a wide adoption of the benchmark. Note that thanks to its design, FLamby’s datasets can be easily plugged in other FL frameworks, allowing researchers to explore privacy-preserving techniques, e.g. cryptographic algorithms.
>
> In order to showcase the extensibility of FLamby, we added an example of differentially private FL with Opacus, see Appendix L.1. In the same fashion, thanks to the simplicity of the aggregation step, see e.g. [here L 148-159](https://github.com/owkin/FLamby/blob/main/flamby/strategies/fed_avg.py), we think it would be very easy for researchers to test cryptography-based secure aggregation method.
>
> ### 5. The scale of the dataset is relatively small, even for a cross-silo setting.
>
> We provide 7 naturally partitioned datasets covering a diversity of tasks, modalities, and dataset sizes (ranging from 96 to 23k), which reflect the reality of healthcare datasets. The relative paucity of medical data compared to other fields (e.g. computer vision) stresses the need for cross-silo FL applications in this field. We believe this work is a step towards this goal.
>
> ### However, the paper lacks details on the evaluation metrics of the benchmark.
>
> We updated the description of the benchmark in the revised version, but we are not sure what the reviewer meant by this remark. We would be happy to further discuss on evaluation metrics details during the discussion period.

---

> ### Author Response · Authors · 2022-08-26
> **Response to Reviewer JW67**
>
> We thank the reviewer again for their feedback and hope we have addressed their questions/concerns. If there are any additional questions, we would be happy to answer them.

---

### Review · Ethics_Reviewer_2iHM · 2022-08-25

**Recommendation:** 1

**Ethics Documentation:**

Yes, all the documentation is well carried out (except for the point I raised in my initial review)

**Ethics Review:**

Having read the initial reviews and the author's rebuttal, I find that the ethical concerns that were raised were adequately addressed by authors both in the paper and in the accompanying repository.

However, one proposal that I have for improving the documentation around datasets is to put the additional ethical details that were added in the paper to the dataset READMEs as well.

For instance, for the TCGA-BRCA dataset, adding the following sentences: "As per the TCGA policies, special care was devoted to ensure privacy protection of research subjects,including but not limited to HIPAA compliance. Note that we do not use the genetic part of TCGA
whose access is restricted due to its sensitivity."  to the dataset [README](https://github.com/owkin/FLamby/blob/main/flamby/datasets/fed_tcga_brca/README.md) can help users understand the limitations of the dataset better (and the same goes for the other datasets, of course).

I would like to congratulate the authors for such a thorough job in addressing the ethical and legal concerns and limitations of their dataset.

---

> ### Author Response · Authors · 2022-08-26
> **Thanks to Ethics reviewer 2iHM**
>
> We would like to thank ethics reviewer 2iHM for their time and review.
>
> Regarding the dataset readmes', we thank the reviewer for the suggestion and updated the documentation accordingly in the online repository.

---

### Author Response · Authors · 2022-08-19
**General answer to reviews**

We thank all reviewers for their time and their feedback. Below we summarize and answer the main review points. We also provide detailed answers to each reviewer in separate comments. We submitted a revised version of the paper taking into account the reviewers’ remarks. We would be happy to further exchange with reviewers during the discussion period.

Among the **strengths** of this work noted by the reviewers, the *diversity of the tasks* covered by the datasets comes often (tsLT, 1pXZ, KtyC). Most reviewers also noted the ease of use of the code’s documented and standardized API (KtyC, hEKK, NALj, tsLT). The *value of this work for the FL community* was also highlighted as a novel contribution in an underexplored area by a majority of reviewers (tsLT, hEKK, JW67, NALj). All reviewers judged the paper *well written*.

A main concern shared by all reviewers is related to the **medical focus of the paper**. Healthcare represents one of the most important application for cross-silo FL. But, it is being held back by lack of public benchmarks. This gap between the existing benchmarks and real world applications leads to wasted research effort improving metrics which may not matter in practice, as well as a lack of confidence in the SOTA methods by practitioners. We believe Flamby will bridge this gap, and can have immense potential impact for the community. We focused on healthcare datasets for now because they have the highest potential impact in the short term. **In order to better reflect this current healthcare focus, we amended the paper (especially the title, abstract and introduction) in the revised version**. However, in the longer term and with the help of the FL community, the dataset suite could be easily extended to other domains.

Regarding privacy, personalization, cluster simulations, we would like to stress that **our focus is not to develop a new comprehensive FL framework, but to bridge the gap between evaluation benchmarks and the real world**. Because of this, we want to focus on being *lightweight, simple, extensible, and compatible with standard frameworks*. We believe this would be critical for widespread adoption of the benchmark by the community. Note that this design choice does not exclude the future extension to account for advanced features like secure aggregation or differential privacy. Indeed, Flamby is designed to  enable future research in cross-silo FL in all its aspects, including privacy (JW67, NALj, KtyC), fairness (KtyC), personalization (tsLT) or benchmarking communication efficiency on real clusters (KtyC). We note that successful benchmarks for cross-device FL, such as LEAF, followed the same approach.

In order to showcase the **extensibility** of FLamby, we added an example of *differentially private FL* with Opacus (Appendix L.1 in revised version) as well as an example of *personalized federated learning* (Appendix L.2 in revised version). In the same fashion, thanks to the simplicity of the aggregation step, see e.g. [here LL 148-159](https://github.com/owkin/FLamby/blob/main/flamby/strategies/fed_avg.py), we think it would be very easy for researchers to test e.g. secure aggregation methods for privacy attacks,  or robust aggregation methods for Byzantine attacks.

Regarding the **compatibility with most FL frameworks** (NALj, 1pXZ), we would like to clarify that this compatibility is made possible through the standardized low-level API introduced by FLamby for each dataset (data loaders, models and losses). The code repository now showcases examples of FLamby datasets with both [FedML and Fed-BioMed](https://github.com/owkin/FLamby/tree/main/integration). This makes it easy for researchers to benchmark their own algorithms of interest on the proposed datasets.

Regarding the **codebase**, we would like to highlight that the code repository has now been made public at this [link](www.github.com/owkin/flamby), following the timeline that was indicated at submission. We plan to develop it further in the future. For existing datasets, we will follow a **maintenance plan** to ensure that issues in the codebase are fixed, and that evolutions in the original datasets (including opted-out patients) are reflected in the dataset suite code, see Appendix B.2 in the revised version. Regarding data storage, the main reason why we limit storing data ourselves is conflicting licenses: FLamby is released under MIT license while several datasets are released under creative commons licenses with share alike clauses.

---

### Meta-Review · Area_Chair_obxR · 2022-09-07

**Recommendation:** Accept
**Confidence:** 5

**Metareview:**

This paper is a great fit for this specific track of NeurIPS. As underlined by the reviewers, it will greatly benefit the federated learning for healthcare community as it provides an holistic starting point containing curated datasets with natural partitioning and baselines. Those elements currently are unfortunately missing for this community, which will be addressed with this article!

---

### Decision · Program_Chairs · 2022-09-16

Accept